# Cannabinoids Transmogrify Cancer Metabolic Phenotype via Epigenetic Reprogramming and a Novel CBD Biased G Protein-Coupled Receptor Signaling Platform

**DOI:** 10.3390/cancers15041030

**Published:** 2023-02-06

**Authors:** David A. Bunsick, Jenna Matsukubo, Myron R. Szewczuk

**Affiliations:** Department of Biomedical and Molecular Sciences, Queen’s University, Kingston, ON K7L3N6, Canada

**Keywords:** health-related diseases, cardiovascular diseases, metabolic disorders, epigenome, epigenetic reprogramming

## Abstract

**Simple Summary:**

The epigenome is dynamic and flexible and can influence its phenotype by turning the genes on and off based on the environment to which it is exposed. Significant epigenetic changes detected are DNA methylation, histone modifications, and RNA-associated alterations. These modifications are potentially heritable across the genome, leading to transgenerational inheritance, and they can even be reversible based on lifestyle modifications. This review focuses on the environmental impact of cannabinoid factors and the subsequent combined effects on epigenetic modifications, such as cancer metabolism, through a biased G protein-coupled receptor signaling paradigm regulating several hallmarks of cancer.

**Abstract:**

The concept of epigenetic reprogramming predicts long-term functional health effects. This reprogramming can be activated by exogenous or endogenous insults, leading to altered healthy and different disease states. The exogenous or endogenous changes that involve developing a roadmap of epigenetic networking, such as drug components on epigenetic imprinting and restoring epigenome patterns laid down during embryonic development, are paramount to establishing youthful cell type and health. This epigenetic landscape is considered one of the hallmarks of cancer. The initiation and progression of cancer are considered to involve epigenetic abnormalities and genetic alterations. Cancer epigenetics have shown extensive reprogramming of every component of the epigenetic machinery in cancer development, including DNA methylation, histone modifications, nucleosome positioning, non-coding RNAs, and microRNA expression. Endocannabinoids are natural lipid molecules whose levels are regulated by specific biosynthetic and degradative enzymes. They bind to and activate two primary cannabinoid receptors, type 1 (CB1) and type 2 (CB2), and together with their metabolizing enzymes, form the endocannabinoid system. This review focuses on the role of cannabinoid receptors CB1 and CB2 signaling in activating numerous receptor tyrosine kinases and Toll-like receptors in the induction of epigenetic landscape alterations in cancer cells, which might transmogrify cancer metabolism and epigenetic reprogramming to a metastatic phenotype. Strategies applied from conception could represent an innovative epigenetic target for preventing and treating human cancer. Here, we describe novel cannabinoid-biased G protein-coupled receptor signaling platforms (GPCR), highlighting putative future perspectives in this field.

## 1. Introduction

‘Epigenetic’ refers to “on top of or in addition to genetics.” A genome is the set of genetic information in the cell’s DNA. Every body cell has the same genome, though their respective phenotypes are different due to the control of gene expression [1]. An epigenome consists of a series of chemical compounds and complex external modifications that can cause changes in the genomic DNA without causing any changes to the DNA sequences of a gene, giving each cell a unique cellular and developmental identity [2]. The epigenome of an organism is dynamic and flexible and can influence its phenotype by turning the genes on and off based on the environment to which it is exposed [1]. Some significant epigenetic changes detected are DNA methylation, histone modifications, and RNA-associated alterations. These modifications are potentially heritable across the genome, leading to transgenerational inheritance. They can even be reversible based on lifestyle modifications [1]. This review focuses on the impact of cannabinoid factors and the subsequent combined effects on epigenetic modifications, such as cancer metabolism, through a biased G protein-coupled receptor signaling paradigm regulating several hallmarks of cancer.

## 2. Cannabis and the Endocannabinoid System

The cannabis sativa plant comprises over 550 chemical compounds [3]. Of these compounds, more than 100 have been identified as phytocannabinoids—plant-derived natural products that act on the endocannabinoid system (ECS) [3]. The most well-studied phytocannabinoids include delta-9 tetrahydrocannabinol (THC), cannabidiol (CBD), and cannabigerol (CBG) [4]. THC is the primary cannabis psychoactive component and is part of the chemical class that constitutes most of the phytocannabinoid content in *C. sativa*. [3,5]. In comparison, CBD is a principal non-psychoactive constituent of cannabis with various pharmacological actions. At the same time, CBG is a minor constituent that serves as the direct precursor to CBD and THC [4]. Endocannabinoids and synthetic cannabinoids represent two other groups of chemical compounds that share similar structures and characteristics to the phytocannabinoids isolated from *C. sativa*. [6]. Endocannabinoids are chemical compounds naturally produced within the human body, while synthetic cannabinoids are produced in the laboratory [6]. The two major endocannabinoids are anandamide (AEA) and 2-arachidonic glycerol (2-AG) [7]. Synthetic cannabinoids represent a large group of chemical compounds, and much research has focused on developing synthetic cannabinoids for medical research purposes and promising therapeutic tools [8,9].

Cannabinoid receptors, belonging to the G-protein coupled receptor (GPCR) family, mediate the biological activity of both endogenous and exogenous cannabinoids [10]. The primary cannabinoid receptors within the body are the CB1 and CB2 receptors [11]. The activation of CB1 and CB2 receptors stimulates cellular signaling via the G alpha subunit (G_i/o_), leading to the inhibition of adenylyl cyclase and the subsequent stimulation of protein kinases that play a crucial role in multiple signaling pathways, including mitogen-activated protein kinase (MAPK) pathways, phosphoinositide 3-kinase (PI3K) pathways, and cyclogeneses (COX) 2 pathways [12]. CB1 is the most abundant GPCR expressed in the mammalian brain [13]. Specifically, the CB1 receptor has high expression in brain regions involved in learning and memory (e.g., hippocampus), motor coordination (e.g., basal ganglia, cerebellum), and cognitive and emotional processes (e.g., amygdala, prefrontal cortex), allowing it to mediate most of the neurobehavioral effects of THC [11,14]. Although a long-standing belief existed that CB2 receptor expression was restricted to peripheral tissues and predominantly in immune cells, low expression of CB2 receptors in the brain has also been reported [15]. CB1 and CB2 receptor expression occurs in tissues outside the central nervous system, including skeletal muscle, liver, immune cells, and reproductive organs [16]. Furthermore, recent studies suggest that other GPCRs besides CB1 and CB2 could also mediate the intracellular effects of cannabinoids, such as GPR18 and GPR55 [17].

Cannabis is widely recognized for its recreational use due to its psychiatric effects, including anxiolysis and euphoria [18]. Moreover, medicinal cannabis is being increasingly explored as a potential therapy to treat various conditions, including chronic pain, vomiting, and nausea due to chemotherapy, depression, anxiety disorder, sleep disorder, and even cancer therapy [19,20]. Despite growing interest in the therapeutic potential of cannabinoids, further research is needed to develop a more comprehensive understanding of the adverse effects of cannabis use [21,22]. Of particular significance are studies that assess the impact of cannabinoids on the tumor microenvironment and epigenetic reprogramming.

## 3. Cannabis and Cancer Metabolism

### Cannabis and the Endocannabinoid System’s Potential as a Cancer Therapeutic

Given the growing interest in cannabis, there is more research on the effects CB1 and CB2 have on tumor growth and metastasis [23]. Several studies have identified a correlation between cannabinoid receptor upregulation, endocannabinoid metabolic enzymes, and ligands in cancerous tissue [24,25,26,27]. If ligand and receptor upregulation is correlated with cancerous tissue, it suggests that the ECS plays a role in cancer modulation. Chakravarti et al., [26] found that cannabinoids mediate cancer cell signaling. However, a study by Velasco et al., [28] demonstrated that *C. sativa* L.’s cannabinoid and non-cannabinoid secondary metabolites have apoptotic, anti-inflammatory, and anti-metastatic properties suggesting cannabis’ properties may be harnessed for anti-cancer therapeutics.

Several in vitro and in vivo studies found that THC and JWH-133, a CB2 agonist, exert anti-proliferative and anti-angiogenic effects on ErbB-2 breast cancer by reducing MMP-2 (Matrix metalloproteinase) and MMP-9 (Matrix Metalloproteinase-9) levels [29,30,31]. Additionally, CBD reduces A549 lung cancer cells through upregulating tissue inhibitors of MMP-1 (TIMP-1) and reducing plasminogen activator inhibitor-1 (PAI-1) [32,33]. JWH-133 inhibits MMP-2 secretion to induce anti-angiogenic properties in A549 cells [34]. Moreover, cannabinoids appear to have anti-proliferative effects on tumors through anti-invasion properties and inhibiting MMPs, impacting cancer metastasis through ECM (Extracellular matrix) degradation [31]. Additionally, the results of some studies suggest cannabinoids may affect EMT (Epithelial-mesenchymal transition) and chemoresistance. For example, the cannabinoid 2-methyl-2′-F-anandamide reduces β-catenin nuclear translocation and transcriptional activity, inhibiting the Wnt/β-catenin pathway [35]. As a result, the expression of β-catenin target genes such as MMP-2, c-Myc, and cyclin D are reduced [35]. Additionally, mesenchymal markers such as N-cadherin, Slug, and Snail are reduced [35]. One study by Caffarel et al., [31] found that a selective inhibitor of the monoacylglycerol lipase (MAGL), a key enzyme in the human endocannabinoid system, that degrades 2-arachidonoylglycerol metabolism, reducing EMT markers and upregulates epithelial markers, including E-cadherin. In breast cancer, the GPCR receptor, GPR55, heterodimerizes with CB receptors, and its targeting impacts tumor growth [36]. Additionally, CB2 heterodimerization with human epidermal growth factor receptor 2 (HER2) and C-X-C chemokine receptor type 4 activates CB2 and inhibits signaling of the aforementioned (word for HER2 and C-X-C), suggesting its potential as an anti-tumor therapeutic [37]. Several studies report that dysregulation of the ECS during carcinogenesis may result in increased cancer aggression [38,39,40,41,42] as the endocannabinoid system is involved in various aspects of tumorigenesis [28,43,44,45]. For example, studies have linked cannabinoids to promoting apoptosis, inducing cell cycle arrest, and having a role in the invasion and self-renewal of tumor cells [28,44,45,46]. In pancreatic cancer, cannabinoids cause apoptosis by inhibiting pancreatic beta cell insulin receptor signaling, which interacts with the CB1 receptor from the MAPK and extracellular signal-regulated kinase (ERK) pathways [47,48]. In breast cancer, the inhibition of the epidermal growth factor (EGF), nuclear factor kappa B (NF-kB), ERK/Akt, and MMP 2 and 9 signaling pathways due to ECS activation results in anti-proliferative effects [49]. The ECS’s ability to inhibit tumorigenesis and other aspects of cancer growth makes it a novel target when treating various cancers. However, more research is needed to define the ECS’ role in tumor heterogeneity and how its signaling impacts the activation of other signaling pathways involved in cancer progression.

## 4. Cannabis and Epigenetics

### 4.1. Cannabis-Induced Epigenetic Effects

Although experimental data about the epigenetic modifications induced by cannabis remain sparse, studies have begun to reveal specific epigenetic modifications associated with cannabis exposure [20]. Modifications of the epigenetic marks such as DNA methylation, histone modifications, and changes in non-coding RNA (ncRNA) regulate gene expression by altering DNA accessibility and chromatin structure [50]. Recently, much research has focused on the association between developmental cannabinoid exposure and aberrant epigenetic markers that could modulate the risk of disease [51]. One study by Schrott et al. demonstrated an association between cannabis use in adult males with significant hypomethylation at 17 CpG sites within the autism candidate gene Discs-Large Associated Protein 2 (*DLGAP2)* in sperm [52]. In a follow-up animal study, the research group demonstrated intergenerational inheritance of this altered DNA methylation pattern, specifically identifying four CpG sites within the *DLGAP2* gene with significant hypomethylation in cannabis-exposed offspring rats compared to controls [52]. Another study by Szutorisz et al. demonstrated that the offspring of parents exposed to THC during their adolescence had an increased propensity to self-administer heroin as adults despite not being exposed to THC themselves [53]. A subsequent investigation of the molecular mechanisms that enable parental THC exposure to influence addictive behavior in offspring revealed 1027 differentially methylated regions in the nucleus accumbens of rats with parental THC exposure [54].

Moreover, one study demonstrated that prenatal cannabis exposure decreased the expression of dopamine receptor D2 (D2R) mRNA—a consistent feature observed in adults with substance use disorders in the human ventral striatum [55,56]. Mechanistic studies in rats revealed that decreased D2R expression in THC-exposed offspring is mediated by epigenetic alterations, specifically by increased dimethylated lysine 9 (2meH3K9) and decreased trimethylated lysine-4 (3meH3K4) on histone H3 [56]. The authors propose that these epigenetic modifications could contribute to addiction vulnerability later in life [56]. Prenatal THC exposure can also lead to the differential expression of micro RNA (miRNAs) [57]. In one study, prenatal THC exposure was associated with the upregulation of miR-122-5p and the resultant downregulation of insulin-like growth factor 1 receptor (Igf1r) [57]. The downregulation of this receptor resulted in an increase in follicular apoptosis among THC-exposed offspring, leading the authors to speculate whether prenatal THC exposure may interfere with critical pathways in the developing ovary, ultimately leading to subfertility [57]. Although studies have begun to reveal associations between cannabis exposure and epigenetic effects, there is a significant gap in knowledge concerning the molecular pathways that mediate the epigenetic modifications induced by cannabis [58].

### 4.2. Molecular Insights into Cannabis-Induced Epigenetic Effects

Current research has only begun to uncover the pathways involved in cannabis-induced changes in DNA methylation, histone modifications, and miRNA expression [59]. Of the existing research, evidence supports the hypothesis that cannabis can alter the activity of histone and DNA-modifying enzymes [58]. For example, Khare et al. demonstrated that THC exposure caused a 3.5-fold increase in histone deacetylase 3 (HDAC3) expression in the BeWo trophoblast cell line [60]. HDAC3 is a component of the nuclear receptor corepressor (N-Cor) repressor complex, responsible for the deacetylation of lysine residues on the *N*-terminal of core histones [61]. Moreover, multiple studies have examined the effect of phytocannabinoids on DNA methyltransferases (DNMT) expression [62,63,64,65,66]. One study compared epigenetic changes in THC-treated and untreated myeloid-derived suppressor cells (MDSCs) [63]. This study revealed an increase in methylation at the promoter regions of DNMT3a and DNMT3b in THC-exposed cells, resulting in a significantly reduced expression of these enzymes [63]. A different study revealed that CBD increased global DNA methylation levels in human keratinocytes by selectively enhancing DNMT1 expression without affecting DNMT3a or DNMT3b expression [62]. Previous studies indicate that phytocannabinoids can modulate the expression of histone and DNA-modifying enzymes; however, the interactions and factors involved in these changes have yet to be elucidated.

Nevertheless, a study by Paradisi et al. investigating the role of the endocannabinoid anandamide in inhibiting human keratinocyte differentiation began to characterize the molecular mechanisms that modulate DNMT activity [67]. The investigators determined that anandamide can decrease the expression of genes involved in keratinocyte differentiation by increasing de novo DNMT activity in a CB1-dependent manner [67]. Most importantly, this study revealed that the interaction between anandamide and CB-1 increases DNMT activity through a p38 and, to a lesser extent, p42/44 MAPK-dependent pathway [67]. Similarly, a more recent study revealed that CBD upregulates the expression of specific genes involved in melanogenesis in a CB1-dependent manner, mediated by the activation of p38 and p42/44 MAPK; however, this study did not investigate the role of DNMTs [68].

Although signaling cascades are often described as isolated entities, biological signaling involves complex interactions between these cascades [69]. For instance, p38 and related kinases can act as cofactors in NF-κB activation, and significant overlap exists between the stimuli that activate NF-κB and the stimuli that activate MAPKs [69]. Interestingly, NF-κB signaling has yet to be studied in cannabinoid-mediated epigenetic effects, even though its association with MAPKs and its ability to induce epigenetic modifications support its involvement [69,70]. The NF-κB family consists of five structurally related transcription factors that form homo- and heterodimers with one another [71]. Inactivated NF-κB dimers are located in the cytoplasm in association with inhibitory proteins called inhibitors of kB (IkB) [72]. A plethora of factors can induce the phosphorylation and subsequent ubiquitin-dependent degradation of IkB, enabling the translocation of its associated NF-κB subunits to the nucleus, where they can regulate the transcription of many genes [72].

Notably, many studies have demonstrated the role of NF-κB in epigenetic regulation [70]. In human acute myeloid leukemia (AML) cells, Tage et al. demonstrated that NF-κB complexes with MUC1-C, a transmembrane oncoprotein at the DNMT1 promoter, drive *DNMT1* transcription [73]. A different study supported these results by demonstrating that the introduction of a proteasome inhibitor known to inhibit NF-κB translocation caused the transcriptional repression of the *DNMT1* gene, leading to the reduced expression of the DNMT1 protein [74]. Ultimately, this induced global DNA hypomethylation, enabling the re-expression of epigenetically silenced genes in AML cells [74].

Furthermore, NF-κB can modulate histone acetylation and methylation [75,76]. Numerous studies have demonstrated that can recruit and position chromatin modifiers, such as acetyltransferases and deacetylases, to the regulatory regions of different genes [70,76,77,78]. Additionally, one study revealed that the direct binding of NF-κB to the promoter of Jumonji domain-containing protein-3 (Jmjd3) is necessary to induce the expression of this histone demethylase [75]. The Jmjd3 demethylase erased the trimethylation marker of lysine 27 on histone H3 (H3K27me3) and was found to induce the expression of specific genes controlling cell fate and differentiation [75]. Another relevant study investigated the involvement of NF-κB in the epigenetic changes induced by the Epstein-Barr virus (EBV) during the transformation of resting B-lymphocytes into permanently growing lymphoblastoid cell lines [79]. This study demonstrated that the NF-κB pathway is essential to EBV-mediated histone methylation changes, including H3K27me3 and H3K4me3 [79]. Additionally, NF-κB was found to bind directly to the transcription start site of upregulated and downregulated miRNAs [79]. In this way, NF-κB directly mediates the expression of miRNAs that control the translation of mRNA transcripts involved in the growth transformation of B-cells [79].

Existing literature confirms the involvement of NF-κB in the mediation of specific epigenetic modifications; thus, it is an exciting candidate that may contribute to the epigenetic changes induced by cannabis. What is intriguing about the effects of cannabis on the NF-κB pathway is that certain cannabinoids act antagonistically to inhibit this pathway while others induce its activation [80]. Generally, studies have found that the interaction of CBD with CB1 and CB2 receptors inhibits the translocation of NF-κB to the nucleus, preventing it from exerting its effects on gene expression [49,81,82,83,84]. In contrast, most studies demonstrate that THC enhances NF-κB translocation to the nucleus [85,86,87,88]. One study showed that treatment with THC induced the phosphorylation of IkB-alpha in dendritic cells in a CB1- and CB2-dependent manner, enabling the release of NF-κB subunits and their translocation to the nucleus [85]. The literature suggests that specific components of cannabis can activate the signaling pathway, which may lead to epigenetic modifications; however, it has yet to be elucidated how these CB1 and CB2 ligands can induce opposing effects on the NF-κB pathway.

Nevertheless, the discovery of biased GPCR agonism may provide unprecedented insights into the opposing effects mediated by cannabinoids [89]. Upon ligand binding, GPCRs undergo a conformational change that leads to the activation of the heterotrimeric G-protein and its dissociation into the G_α_ subunit and G_β/y_ heterodimer [90]. These subunits interact with many effectors, leading to complex G-protein-mediated signaling pathways [90]. Biased GPCR agonism refers to the ability of GPCR ligands to activate distinct intracellular signaling pathways by preferentially stabilizing different active conformational states of the receptor [91]. Both orthosteric ligands, such as THC, and allosteric ligands, such as CBD, tend to induce biased signaling at GPCRs [91]. Importantly, biased agonism and allosteric modulation have been observed at the CB1 receptor; however, to our knowledge, biased modulation of NF-κB signaling has yet to be studied in the context of CB receptors [91]. Cannabinoids may induce epigenetic modifications through the activation of these glycosylated receptors and the subsequent initiation of NF-κB; however, cannabinoids do not activate these receptors directly [92]. Rather, the binding of cannabinoids to CB receptors, as depicted in Figure 1, can stimulate the activity of RTKs, TNFRs, and TLRs through a process known as transactivation [93,94,95].

Matrix metalloproteinase-9 plays a crucial role in the remodeling of the extracellular matrix. With the novel discovery of receptor transactivation or GPCR agonist-bias signaling, MMP-9 has been shown to play a more prominent role in ECM remodeling. Through activation of associated GPCRs, such as the angiotensin AT1R and bradykinin receptors, MMP-9 is activated to induce activation of RTKs such as the TrkA receptor, EGFR, and IR as well as Toll-like receptors in the complex with NMBR, IR, and Neu1 in naïve (unstimulated) and stimulated RAW-blue macrophage cells. Here, a molecular link regulating the interaction and signaling mechanism(s) between these molecules on the cell surface uncovers a biased GPCR agonist-induced cell surface and intracellular Toll-like receptor (TLR) transactivation-signaling axis, mediated by Neu1 sialidase and the glycosylation modification of TLRs. The biased GPCR-signaling platform here potentiates Neu1 and MMP-9 cross-talk on the cell surface, which is essential for the transactivation of TLRs and subsequent cellular signaling.

The initiation of NF-κB signaling can begin with the activation of a variety of glycosylated transmembrane receptors, including receptor tyrosine kinases (RTKs), receptors for tumor necrosis factor (TNFR), and Toll-like receptors (TLRs) [101,102,103]. The signaling pathways induced by these receptors can all lead to the phosphorylation of IkBα, enabling the translocation of NF-κB to the nucleus [101]. Cannabinoids may induce epigenetic modifications by activating these glycosylated receptors and initiating NF-kB; however, cannabinoids do not activate these receptors directly [92]. Rather, the binding of cannabinoids to CB receptors can stimulate the activity of RTKs, TNFRs, and TLRs through a process known as transactivation, as depicted in Figure 1 [94,95,104].

The factors controlling the transactivation of glycosylated receptors by CB receptor agonists remain poorly understood. To this end, the biased NMBR-MMP9-Neu1 signaling axis highlighted throughout this review may be essential to transacting RTKs and TLRs by CB receptor ligands. A series of published studies demonstrated the involvement of this signaling axis in the transactivation of various receptors, including the insulin receptor [105], Trk receptors [106], and Toll-like receptors [107]. Notably, one study demonstrated that the GPCR agonist bombesin could induce TLR4 transactivation via this signaling paradigm, leading to the activation of NF-κB in BMC-2 and RAW-blue macrophage cells [107].

### 4.3. Future Directions of Cannabis-Induced Epigenetic Rewiring Research

The literature is conclusive that cannabis can evoke epigenetic modifications; however, the research on the pathways that mediate these epigenetic changes is still in its infancy [58]. A promising candidate for mediating the epigenetic effects induced by cannabis is NF-κB [108]. Existing literature demonstrates that certain cannabinoids, including THC and WIN 55, 212-2, enhance the activation of this pathway, while others, such as CBD, suppress its activation [80,87,109,110]. This finding supports the involvement of a ligand-biased signaling paradigm [91]. Specifically, the biased GPCR-Neu1-MMP9 signaling axis, as depicted in Figure 2, may control the transactivation of RTKs by the cannabinoid receptors, ultimately leading to the downstream activation of NF-κB and the induction of epigenetic reprogramming. Interestingly, Rozenfeld et al., [111] found heteromers of type I cannabinoid GPCR receptor (CB1R) with angiotensin II receptor type 1 (AT1R) mediating enhancement of angiotensin II-mediated signaling. Haxho et al., [105] found that angiotensin II GPCR receptor type I forms heteromers with NMBR in a multimeric receptor complex with Neu1, IRβ, and NMBR in naïve (unstimulated) and stimulated HTC-IR cells with insulin, bradykinin, angiotensin I and angiotensin II (Figure 2).

These innovative research findings have the potential to elucidate an entirely new signaling process for these receptors and cell responses. To gain a deeper understanding of how cannabinoids evoke epigenetic modifications, we must ascertain a more comprehensive understanding of the activity of the cannabinoid receptor itself and its interactions at the cell surface.

## 5. Neuraminidase-1

### 5.1. The Neuraminidase-1 Complex

Neuraminidase-1 (Neu-1) is a sialidase expressed in all mammalian tissues and is active on sialylated glycoproteins [114]. Recent research has found that Neu-1 is involved in a nerve growth factor (NGF)-induced TrkA receptor involving neuromedin B (NMBR) GPCR-signaling process in complex with MMP-9 to form a multimeric enzymatic axis with Neu-1 sialidase, which has a critical role in ligand-induced activation of RTKs and TLRs [115,116]. NMBR-MMP-9-Neu-1 cross-talk occurs when ligand-binding its receptor such as epidermal growth factor receptors (EGFR) [106], nerve growth factor TrkA [116], and insulin receptor (IR) [112]. Upon binding to an RTK or TLR, a ligand-dependent GPCR signaling occurs where NMBR GPCR signaling through a Gαi-protein subunit activates MMP-9 to induce Neu-1 activity [117]. The activation of Neu-1 is mediated by MMP-9, which removes the elastic binding protein (EBP) in complex with both Neu-1 and a protective protein cathepsin A. Neu-1 hydrolyzes α-2,3-sialic acid residues on glycosylated receptors to allow receptor activation and dimerization by removing steric hindrance [118]. Upon activation, downstream signaling occurs, resulting in conditions such as tumorigenesis and insulin resistance (Figure 1), which will be discussed in this review [112,115,116].

### 5.2. Neuraminidase-1′s Impact on Cancer and Diabetes Progression

There are many cellular mechanisms by which cancer and diabetes may occur. However, recently, glycosylation has been a critical factor in cancer metastasis. A review by Vajaria et al., [119] discussed that the glycosylation of proteins on a cell surface and the loss of E-cadherin due to MUC alterations has a significant role in tumor metastasis as it contributes to metastasis, angiogenesis, proliferation, and growth suppressor evasion. Moreover, since glycosylation plays an essential role in cancer, therapies targeting the mechanisms of these receptors’ glycosylation would likely have significant therapeutic potential.

The IR, a highly glycosylated receptor, has 18 glycosylation sites [120] and has implications for insulin resistance, diabetes, and cancer if the insulin-induced IR activation is not functioning correctly [121,122,123]. A study by Arabkhari et al. determined that when rat skeletal L6 myoblast cells were treated with Neu1 sialidase, desialylation of IR occurred, resulting in an increased proliferative response to lower insulin doses [124]. These findings indicate that glycosylation of the insulin receptor increases the receptor’s sensitivity when the ligand binds. Previous studies identify a novel cross-talk between Neu-1 and MMP-9 along with the neuromedin B GPCR, showing that Neu-1, MMP-9, and neuromedin B GPCR (NMBR) form a complex with IRβ subunits on a cell’s surface as neuromedin B is critical for insulin-induced IR activation [112]. The study’s results highlight that Neu-1 activation plays a role in the activation of the IR, supporting Fischoeder et al.’s findings that insulin mediates increases in MMP-9 via IR activation [125]. These findings highlight Neu-1′s importance in IR signaling and how overexpression of Neu-1 may result in overexpression of IR, therefore causing insulin resistance, diabetes, and cancer.

The EGFR is an ErbB RTK receptor that is a critical pathway involved in cell growth, regulation, proliferation, and differentiation of cells [126]. The EGFR is activated along with cytoplasmic tyrosine kinase activity when the EGF binds to the extracellular domain of the EGFR [127]. It is well known that EGFR is often mutated or overexpressed in various cancer types and is a target for anti-cancer therapies [128]. Moreover, the receptor exists as a single, highly glycosylated subunit, making it susceptible to overexpression and cancer. A study by Gilmour et al. found Neu-1 and MMP-9 cross-talk with EGF receptors on EGFR-expressing cells and human pancreatic MiaPaCa-2 and PANC-1 cancer cells, which is critical for EGF activation of EGFRs and eventual cellular signaling [116]. The cross-talk likely occurs from EGFR’s activation of NMBR as it activates the Gαi-protein, triggering the activation of MMP-9 and the removal of the EBP, which activates Neu-1 [129]. The activated EGFRs can then promote tumorigenesis by activating the JAK/STAT (Janus kinase/signal transducers and activators of transcription), PI3K/Akt, and MAPK signaling pathways [106,130,131].

Neu-1′s involvement in regulating glycosylated growth factors receptors such as IR, EGFR, and TrkA indicates it impacts a tumor’s characteristics, such as cell growth, proliferation, and differentiation [106]. Moreover, Neu-1, a regulator of these receptors, is a novel targeted approach for cancer therapeutics. Haxho et al., [132] reported a review on the role of mammalian Neu1 in complex with MMP-9 and G protein-coupled receptors tethered to RTKs and TLRs as a major target in multistage tumorigenesis. The therapeutic efficacy of targeting Neu1 may disrupt several molecular signaling pathways. Given the ability of oseltamivir phosphate (OP) to increase E-cadherin expression and decrease N-cadherin and VE-cadherin expression, as previously reported [133], tumors treated with this OP drug may become more adherent to the surrounding tissue and not metastasize. Haxho et al., [132] proposed a graphical illustration (Figure 3) depicting a Snail-MMP9 signaling axis [134], maintaining several important cancer growth factor receptor signaling platforms in promoting Neu1-MMP9 cross-talk in complex with glycosylated receptors. OP treatment strategies under dose dependence would have a horizontal targeted approach, in which different oncogenic signaling pathways involved in tumorigenesis are targeted with promising therapeutic intent.

### 5.3. Inhibition of the Neu-1 Complex as a Potential Therapeutic to Cancer

Cancer treatments aim to modulate cellular pathways essential for cancer survival and growth, such as RAS, EGFR, vascular endothelial growth factor VEGF, and MMPs. Moreover, identifying and inhibiting complexes influencing these pathways, such as Neu-1 and MMP-9, is a beneficial approach to decreasing tumorigenesis [135]. An essential aspect of utilizing therapies that target different molecular pathways is that the risk of drug resistance is mitigated [136,137,138]. Oseltamivir phosphate (OP), also known pro-drug of Tamiflu, is a licensed medication that targets the neuraminidase protein [139]. Many studies have highlighted the efficacy and therapeutic potential of OP on various types of cancers, such as breast and pancreatic cancer [116,133,140]. Studies performed by Amith et al., [141] and Abdulkhalek et al., [142] found that OP specifically targeted and inhibited Neu-1 activity. Following Neu-1 inhibition, OP exerted its anti-cancer effects through increased E-cadherin expression and decreased VE-cadherin and N-cadherin in human pancreatic cells (PANC1) as EMT and E-cadherin loss [133]. Cell-to-cell adhesion promotes metastasis and tumor progression [133]. OP inhibition of Neu-1 inhibition was also found to downregulate EGFR pathways such as the JAK/STAT, PI3K/Akt, and MAPK [133], all of which are involved in tumorigenesis [106,130,131].

A significant issue regarding cancer therapies is the occurrence of chemo and drug resistance. A study performed by Zhang et al., [143] repurposed the drug aspirin (ASA) as it is an inhibitor of NF-κB signaling, which is involved in the upregulation of MMP-9 [144,145]. The study found that ASA increases the efficacy of gemcitabine, a chemotherapy medication, by inhibiting inflammatory proteins such as NF-κB and reducing the self-renewal potential of cancer stem-like cells (CSCs). They also found that ASA has a preventive effect on the development of pancreatic ductal adenocarcinoma (PDA) and targets highly aggressive PDA cells [143], indicating that ASA has strong therapeutic potential in treating cancer. A recent discovery found that ASA and celecoxib, a selective COX-2 inhibitor, dose-dependently inhibit EGF-induced sialidase activity in PANC1 and MiaPaCa-2 cells through the inhibition of Neu-1 cleavage of the α-2,3-sialic acid [146]. Using this information as a rationale, Sambi et al., [147] utilized the triple combination of ASA, metformin (Met), and OP treatment with tamoxifen on breast cancer cell lines. They found that the cocktail reduced cell proliferation, upended tamoxifen chemoresistance, and increased apoptotic activity in MDA-MB-231 triple-negative breast cancer cells and their tamoxifen-resistant variant. These results suggest a therapeutic approach to multistage tumorigenesis in triple-negative breast cancer.

Recently, Qorri et al., [148,149] investigated the therapeutic potential of OP, ASA, and the chemotherapeutic agent gemcitabine (GEM) in pancreatic ductal adenocarcinoma cancer (PDAC) cells. They found that ASA+OP+GEM upended MiaPaCa-2 and PANC-1 pancreatic cell survival mechanisms, including viability, promoting apoptosis, and expressing extra-cellular matrix proteins. Moreover, it is a novel approach to targeting the survival mechanisms of pancreatic cancer metastasis.

In preclinical studies, Qorri et al., [149] reported that a continuous therapeutic targeting of Neu-1 using parenteral perfusion of oseltamivir phosphate (OP) and aspirin (ASA) with gemcitabine (GEM) treatment significantly disrupted tumor progression, critical compensatory signaling mechanisms, EMT program, cancer stem cells (CSC), and metastases in a preclinical mouse model of human pancreatic cancer. In addition, they demonstrated that ASA- and OP-treated xenotumors significantly inhibited the metastatic potential when transferred into animals.

## 6. MMP-9 as a Therapeutic Target of Cancer

### 6.1. MMP-9 and Its Interaction with the Neu-1 Complex

As mentioned previously, MMP-9 has an essential role with Neu-1 as it forms a trimeric complex involved in ligand-induced activation of RTKs [115,116]. Snail’s transcription factor mediates ovarian tumor neovascularization and is closely associated with MMP-9 expressions in invasive tumors due to similarities in extra-cellular matrix remodeling during invasive processes [134,150]. Jorda et al., [151] found that Snail can up-regulate the expression of MMP-9 by activating the MAPK and PI3K signaling pathways, indicating a link between them. The connection suggests that the Snail-MMP-9 signaling axis is the missing component in promoting growth factor glycosylation modification that results in Neu-1-MMP-9 cross-talk at the ectodomain of RTKs [134]. From there, downstream signaling occurs, involving various pathological conditions, including tumorigenesis and insulin resistance, making MMP-9 a viable target for cancer and diabetes therapies [112,115,116].

### 6.2. Inhibition of the NF-κB Pathway Results in Downregulation of MMP-9

NF-κB is a transcription factor that regulates innate and adaptive immune functions. Moreover, its activation induces pro-inflammatory gene expression, such as cytokines and chemokines [71]. NF-κB activates in response to infectious agents and pro-inflammatory cytokines from the IκB kinase complex. Studies have found that the deletion of the inhibitor kappa-B kinaseβ (IKKβ) in epithelial cells decreases tumor incidence, and the deletion of IKKβ in myeloid cells decreases tumor size, indicating that inactivating the IKK/NF-κB alters the formation of inflammation-associated tumors [152]. Moreover, the NF-κB pathway has a strong association with cancer. Various studies have shown that the promoter region of the MMP-9 gene includes binding sites for NF-κB and activator protein-1. MMP-9 expression requires both transcription factors to be present [153,154,155,156,157]. A study inhibiting the expression of NF-κB in CT26 colon cancer cells through the overexpression of the IκB-α super-repressor found a decrease in MMP-9 expression, highlighting a link between the two [158].

Cancer, a metabolic disease, often has metabolic alterations in cancerous cells, such as an increase in glycolytic metabolism rate known as the “Warburg effect” [159]. Interestingly, therapies such as the ketogenic diet (KD) have been developed to create a metabolic shift towards fatty acid utilization and develop antitumor activity [160], as recent research has shown that a KD may inhibit tumor progression by altering the activity of inflammatory responses [161]. A study by Zhang et al., [162] examined the impact of the limited carbohydrate KD on growth and apoptosis in CT26+ colon cancer cells and the role of protein expression, such as the NF-κB pathway. They found that KD treatment slows colon cancer progression by significantly reducing the protein expression of NF-κB 65, phosphor NF-κB p65, and MMP-9. The results indicate KD downregulates phosphor NF-κB expression, inhibiting the protein levels of their downstream targets such as MMP-9 [162].

Nitric oxide (NO) is also a regulator of MMP-9 activation [163,164,165,166,167]. In-vitro studies show murine macrophages secreting increased levels of MMP-9 during exposure to low NO concentrations due to suppressing the tissue inhibitor of metalloproteinase 1. However, increasing NO concentrations in guanylyl-cyclase-dependent and independent signaling pathways decreased MMP-9 activity [166]. Several studies have observed that the differences in MMP-9 expression from varying NO treatments also involve the regulation of NF-κB as its activation increases or decreases depending on low or high concentrations of NO [167,168], suggesting NO affects NF-κB function. Reports indicate that opioid agonists can induce apoptosis in cancer cells and inhibit angiogenesis [168,169,170,171]. Morphine, an opiate, has been shown to impact tumor progression from its effects on MMP-9 expression and increase NO production [169,170,172,173,174]. A study by Roy et al., [175] found that morphine modulates NF-κB transcription factor and has opioid functions in various cells, indicating morphine regulates MMP-9 production by interfering with NF-κB binding to the promoter region of MMP-9 via opioid receptor-mediated and NO-dependent mechanisms [175,176]. Moreover, morphine as an anti-cancer drug through MMP-9 inhibition shows therapeutic potential. However, despite determining that morphine’s anti-cancer properties, its activation and inhibition vary with cancer and cell types, the dose of morphine, and other experimental factors, suggesting more information is needed on the mechanisms of these receptors and pathways [173,177,178,179,180,181,182].

Epigenetics is the interaction between the genome and the environment as changes to one’s environment result in modifications such as DNA methylation, histone modification, and altering chromatin structure, resulting in changes in gene expression [50]. These changes have been thought to have the potential to silence genes promoting cancer, making epigenetics a promising area of research for novel cancer therapeutics [183]. A study by Sung et al., [184] found that anacardic acid inhibits the acetylation of p65, a subunit of NF-κB, by suppressing histone acetyltransferase (HAT) activity, thereby impeding TNF induced NF-κB activation. The suppression of HAT activity prevents the addition of acetyl groups which typically promote gene expression. They also found that anacardic acid decreases MMP-9 activation due to NF-κB being unable to bind to the promoter of MMP-9. These results are supported by other studies [185] and prove that inhibition of the NF-κB pathway decreases the expression of MMP-9 and suggests epigenetic changes may be used in cancer therapeutics. MicroRNAs, another epigenetic modulator, are single-stranded, non-coding sequences of RNA that are involved in gene expression regulation at the post-transcriptional and translational levels [186,187]. NF-κB regulates four miRNAs, miR-10b, miR-17, miR-21, and miR-9, which are involved in tumor growth and metastasis [188,189,190,191]. Using the luciferase reporter assay, Li et al., [192] found that p65 NF-κB directly regulates miR-10b, miR-17, miR-21, and miR-9 and that the NSAID, sulindac (SS) inhibits the nuclear translocation of NF-κB by decreasing the phosphorylation of IKKβ and IκB. As a result, SS inhibits tumor cell invasion by hindering the NF-κB pathway and miRNAs involved in tumorigenesis. SS inhibiting NF-κB increases the likelihood that MMP-9 is decreased as reduced NF-κB expression is associated with a reduction in MMP-9. However, research needs to be conducted to determine this hypothesis.

### 6.3. Inhibition of the COX-2 Decreases MMP-9 Activity

Cyclooxygenase (COX) -1 and -2 are the most well-known oxygenases involved in catalyzing molecular oxygen into organic compounds. COX-1 is typically expressed in most tissues, whereas COX-2 responds to cytokines and growth factors. They catalyze the reduction of arachidonic acid to prostaglandin G2 (PG) and PGE2 [193]. PGH2 often undergoes metabolism by enzymes to form PGs, which exert their effects through GPCR mechanisms [194]. Research has found that several cancers, including breast, colon, and pancreatic cancers, have increased levels of PG [195,196,197]. Additionally, a 2001 study by Surh et al., [197] found that the NF-κB pathway modulates the expression of COX-2, resulting in metastasis and tumorigenesis, highlighting a link between the NF-κB, COX-2, and cancer. Several studies have found that MMP-9, COX-2, and VEGF expressions exist in many tumor tissues [198,199]. One study found that VEGF expression in colon cancer increases following transfection with COX-2 and that NS398, a COX-2 inhibitor, reduces its effect, indicating COX-2 stimulates tumor angiogenesis through the VEGF pathway [200]. Studies by Larkins et al., [201] and others [202] support the previous findings as they found that selective inhibitors of COX-2 down-regulate MMP-9, -2, and VEGF expression, thereby reducing head and neck squamous cell viability. Moreover, the inhibition of COX-2 is a promising therapeutic in inhibiting MMP-9 and the Neu-1 complex.

The non-steroidal anti-inflammatory drug (NSAID) aspirin inhibits COX-1 and -2 isoforms, exerting anti-inflammatory, antipyretic, and analgesic effects. However, the exact mechanism needs to be studied further, but COX-dependent and independent mechanisms have been proposed [203]. One COX-dependent mechanism proposed is that ASA inhibits COX-2, resulting in the downregulation of phosphatidylinositol 3-kinase (PI3K) signaling [204]. The PI3K pathway is involved in MMP-9 expression [205], which explains why regular ASA use following a colorectal diagnosis increases survival among patients with PIK3CA tumors [204]. ASA also acts through COX-independent pathways to inhibit IKKβ, preventing NF-κB activation and contributing to ASA’s anti-cancer effects [205]. A study by Xue et al., [206] found that aspirin decreases COX-2 mRNA expression, thereby reducing MMP-9 content in macrophages derived from THP-1 cells due to a decrease in mPGES-1 mRNA. They verified that MMP-9 formation was through the COX-2/mPGES-1 pathway when MMP-9 expression increased following increases in mPGES-1 production due to PGE2 stimulation [205].

Curcumin, a naturally occurring polyphenol derived from turmeric, demonstrates anti-cancer properties through modulating various molecular targets such as STAT3, EGFR, PI3KAkt/mTOR, E-cadherin, MMPs, and COX-2 [207]. Curcumin downregulates COX-2 and EGFR expressions since PGE2 transactivates the EGFR pathway to promote cancer cell motility, indicating a cross-talk between the two pathways. The association between the two pathways is a reduction in extracellular signal-regulated kinase (ERK) activity, thereby increasing apoptosis and decreasing cell survival [208]. Again, COX-2 downregulation through curcumin is associated with inhibiting the NF-κB pathway. Here curcumin suppresses IKK, inhibiting phosphorylation and degradation of IKB-kinase alpha and nuclear translocation of p65 [209]. MMP-9 protein levels decreased following curcumin treatment through downregulating ERK signaling pathways that have been established to involve COX-2 [210]. However, not every cancer cell contains COX-2, as studies have found that MiaPa-Ca-2 pancreatic cancer cells have absent levels of COX-1 and -2, and PANC-1 cells have high COX-1 expression and little COX-2 expression [148]. The reduction of COX in pancreatic cancer cells does create problems regarding treatment as the use of NSAIDs becomes cell and dose-dependent to ensure proper treatment.

Additionally, more research can be conducted on the mechanisms of COX on cancers to help determine the exact mechanisms of COX inhibitors such as ASA and celecoxib [146]. However, these findings support COX-2′s role in the Neu-1 complex through its ability to inhibit MMP-9 through the NF-κB pathway. Moreover, COX-2′s role in MMP-9 inhibition indicates its potential as an anti-cancer therapeutic.

## 7. Effects of Biased GPCR Agonists on the Insulin Receptor (IR)

### Bias GPCR Agonism and the Insulin Receptor

The IR is a tyrosine kinase transmembrane signaling protein that regulates cell differentiation, growth, and metabolism. The IR is a ligand-activated receptor whose primary function is metabolic regulation [211]. The receptor comprises two α and two β subunits linked through disulfide bridges [212,213,214]. Insulin receptor binding induces a conformational change, activating IR-β kinase activity [215]. As a result, the downstream effects of IR signaling occur, such as glucose homeostasis for various cell types [216,217]. However, insulin resistance may occur when normal or increased insulin levels produce a reduced biological response, usually when there is reduced insulin sensitivity [218,219]. Although it is suggested that insulin resistance occurs due to the downregulation of the IR, or insulin receptor substrate (IRS) proteins, the precise mechanism of insulin resistance is unknown [123,220]. Insulin resistance has clinical implications in many pathological conditions, including cancer, type 2 diabetes (metabolic syndrome), cardiovascular disease, hypertension, and polycystic ovary syndrome (PCOS) [220]. Blaise et al., [221] found that κ-elastin (kE) exposure decreases sialic activity on the β-chain of the IR, suggesting that Neu-1 desialylates the IR. This study’s results confirm the findings from others [222] that Neu-1 has a significant role in IR regulation, as mice with a Neu-1 deficiency develop hyperglycemia and insulin resistance twice as fast as the wild-type when exposed to a high-fat diet.

Moreover, a neuromedin B GPCR-MMP-9-Neu-1 complex is tethered to IRβ subunits, indicating that MMP-9-Neu-1 cross-talk occurs after insulin binding to its receptor initiates GPCR signaling/MMP-9 activation to induce Neu-1 [112]. Evidence of this was found in a study by Schlien et al., [223], where insulin binding to its receptor induces a conformational change at the *C*-terminus domain. Given that GPCR agonists positively regulate the insulin receptor, and MMP-9 and Neu-1 are associated with the IR, biased GPCR agonism of these GPCRs likely affects the receptor. A study by Haxho et al., [105] found that bradykinin, angiotensin I, and II dose-dependently induce sialidase activity in HTC-IR cells, indicating they exist in a multimeric receptor complex with Neu-1, IRβ, and NMBR. These findings confer that Neu-1-mediated desialylation of the IRβ subunits transactivates insulin receptors following GPCR agonism and IR glycosylation. Moreover, the results confirm findings from previous literature [112] of NMBR’s regulatory role in inducing Neu-1 sialidase and MMP-9 cross-talk for IRβ desialylation and receptor activation. Studies show that IR and GPCR cross-talk regulate various cellular processes, and the dysfunction between the two receptors results in diseases such as metabolic syndrome, type 2 diabetes, and cancer [224]. The dysfunction between the receptors can be caused by GPCR agonists, which can positively regulate IR signaling despite the absence of insulin [105]. A study by Isami et al., [225] confirms Haxho et al.’s, [105] findings that GPCR agonism mediated by Neu-1 transactivates the IR without insulin, possibly enabling various pathological diseases.

## 8. Bias GPCR Agonism on the IR and Cancer

Many studies have implicated that the insulin-like growth factor (IGF) and insulin signaling pathways influence carcinogenesis and tumor progression [226,227]. IGF-1 binds to both the IGF-1 receptor (IGF1R) and IR and, as a result, promotes mitogenic signaling events, inhibits apoptosis, and increases cell proliferation [227]. For example, analyses of specific IR ELISAs found that 80% of breast cancers have significantly higher IR expression than normal breast tissue. The study also found that 20% of the cancer tissue had IR expression ten times the mean value of normal breast tissue [226]. IR overexpression in cancers makes the IGF1R, IR, and the pathways they activate promising therapeutic targets [228,229,230]. For example, when IGF-1 binds to IGF1R and IR, the PI3K-AKT-TOR and RAF-MAPK pathways activate, promoting cell survival and proliferation [227,231]. These pathways are also involved in MMP-9/Neu-1 cross-talk, highlighting that transactivation of the IR is likely mediated by Neu-1 to promote signaling pathways involved in tumorigenesis, and its downregulation will likely lead to a decrease in cancer viability [231,232,233]. Further evidence of Neu-1′s involvement in IR signaling is that in-vitro and in-vivo studies have concluded that insulin affects tumor progression by acting on the IR and not solely by IGF-1R cross-talk [234,235]. GPCR agonists support these conclusions by indirectly transactivating insulin receptors via Neu-1 mediated desialylation of IRβ subunits without insulin, highlighting that the IR plays a critical role in maintaining the vitality of cancer cells and creates a therapeutic target for the treatment of these diseases [224].

Experimental studies have shown that OP and NMBR inhibitor BIM-23127 inhibits independent autophosphorylation of IRβ and IRS-1 in HTC-IR cells, blocking Neu-1-mediated transactivation of the IR, which may be used as a therapeutic target for cancer [112]. A study by Rozengurt et al., [235] found that the mammalian target of rapamycin (mTOR) complex 1 (mTORC1) is critical for IR/IGF-1R and GPCR signaling. Metformin, an antidiabetic drug, induces AMP kinase activation, negatively regulating mTORC1 and suppressing the growth of PANC1 cells [236,237], showing that IGF-1R disruption decreases tumorigenesis as IR/IGF-1R cross-talk is disabled. Linsitinib, a small-molecule tyrosine kinase inhibitor (TKI), blocks autophosphorylation of IGF1R/IR and exerts promising anti-tumor activity [238]. However, clinical trials evaluating its efficacy in metastatic breast cancer found that toxicities such as hyperglycemia would occur as IR disruption would impact proper glucose homeostasis [239]. Similarly, BMS-754807, an ATP-competitive TKI of IGF1R and IR, inhibits pancreatic cancer cell lines. However, like Linsitinib, clinical trials were halted following patients reporting side effects such as impaired glucose tolerance, fatigue, and nausea likely due to excessive IR inhibition [240]. Although discouraging, these results promise that proper targeting of the IR and IGF-1R will inhibit cancer cell vitality and growth. Bromo- and extra-terminal domain (BET) proteins, such as BRD2, BRD3, BRD4, and BRDT, are epigenetic regulators that connect chromatin modifications to the transcriptional activation of genes. They have a critical role in homeostasis, and cell survival, as their dysfunction plays a role in cancer. BET proteins regulate EMT (Snail), which has a significant role in MMP-9 expression and Neu-1 activation [240]. BET protein action and the IGF system are connected as the IGF1R gene directly targets BRD4 in various cancer cell lines. Treatment of RKO-1, PC-3, and MCCL-357 cells with BET inhibitors JQ1 or MS417 reduces BRD4 binding to IGF1R, reducing mRNA expression, and inhibiting the PI3K/Akt pathway [241]. PI3K and BET inhibition hinders the PI3K pathway, thereby reducing cell growth and vitality in cell models [241]. However, more research must be conducted on BET inhibition and MMP-9/Neu-1 activation.

The insulin receptor substrate (IRS) proteins play a pivotal role in metabolism and cancer. Under normal physiological conditions, these proteins are involved in IR/IGF-1R-mediated metabolic regulation by amplifying the PI3K signaling pathway to activate the serine-threonine kinase AKT [242,243]. Evidence of this is that IRS proteins have multiple PI3K binding motifs to recruit and activate PI3K via SH2-domains [244]. Additionally, IRS proteins are signaling intermediates in insulin-regulated glucose homeostasis by promoting glucose uptake [245]. However, dysregulation and overexpression of these proteins lead to pathological conditions such as cancer and diabetes. Many studies have found that IRS-1 negatively regulates tumor progression, whereas IRS-2 promotes tumor behavior through metabolism regulation [246]. As a result, targeting these proteins may have strong potential in novel cancer therapeutics. A study by Zhao et al., [247] found that miR-766, a miRNA, directly targets IRS2 and inhibits activation of the PI3K/Akt pathway to reduce papillary thyroid cancer (PTC) malignancy. However, a limitation of the study is that the association between miR-766 and the prognosis of PTC patients was not analyzed, making its clinical implications unclear. Another study found that long noncoding adipogenesis regulatory factor-antisense RNA 1 (ADIRF-AS1) is overexpressed in osteosarcoma (OS).

Moreover, ADIRD-AS1 impedes tumor growth and is a competitive endogenous RNA for miR-761 as it is removed from IRS1, leading to increased IRS1 overexpression. The study’s results suggest that reducing miR-761 in OS cells neutralizes the effect of ADIRF-AS1 as IRS1 is restored and no longer overexpressed [248]. These two studies highlight the therapeutic potential of miRNAs and how their use may be efficacious in reducing IR expression in cancer [159,160]. Another miRNA, miR-30e, is downregulated in many cancer cell lines [249,250]. A study by Liu et al., [251] found that miR-30e recognizes the 3′-UTR of IRS1 transcripts, indicating that miR-30e expression impacts IRS1 expression. Evidence of this is that cells expressing miR-30e abolish IRS1 in breast cancer cells, thereby suppressing breast cancer cell growth [251]. A preclinical study by Garofalo et al., [252] supports the evidence of inhibiting IRS-1 and -2 in treating cancer. They found that NT157, a selective inhibitor of IRS-1 and -2, decreases the expression of these proteins and downregulates IGF-1R-mediated AKT activation. As a result, NT157 treatment leads to cell cycle arrest, apoptosis, and reduced cell motility in OS cells, thereby reducing OS metastasis [252].

## 9. Conclusions

Here, we reviewed reports demonstrating that GPCR agonists bombesin, bradykinin, angiotensin I, and angiotensin II, each significantly and dose-dependently induced sialidase activity in live IR [105,112], EGFR [116], Trk [106], and TLR [107,115,141,142] expressing cells, in vitro. If Neu1 sialidase activity is required for these glycosylated receptor activations in a mechanism dependent on GPCR Gαi signaling and subsequent MMP9 activation, this led us to hypothesize that GPCR agonists may indirectly transactivate receptors via Neu1-mediated desialylation process and downstream signaling. The reported data support this hypothesis. Furthermore, GPCR agonist-induced sialidase activity in these cells resulted in receptor ligand-independent signaling in these cells. This process is blocked by Neu1 inhibitor oseltamivir phosphate and the neuromedin B GPCR (NMBR) inhibitor BIM-23127. These findings are consistent with our previous report describing the regulatory role of NMBR in inducing Neu1 sialidase and MMP9 cross-talk required for IRβ desialylation and receptor activation [112]. The capability of GPCR agonists to positively regulate glycosylated receptor signaling in the absence of their ligands presents a role of cannabinoid receptors CB1 and CB2 signaling in activating numerous receptor tyrosine kinases and Toll-like receptors in the induction of altered epigenetic landscape(s) in cancer cells which might transmogrify cancer metabolism and epigenetic reprogramming to a metastatic phenotype.

## 10. Future Research Directions and Limitations

The gaps in our knowledge regarding how CB cannabinoids are implicated in host metabolism via GPCR receptors reinforce the importance of research needed to understand the mechanistic action of these drugs on the body as promising metabolic candidates. The specific mechanisms and details of the effects of CB cannabinoids on host metabolism and energy homeostasis still remain to be elucidated. It is likely that future research in the area of cannabinoid pharmacology will be directed at:Exploring the structure-activity relationships of ligands that target the CB1 allosteric site or that behave as neutral CB1 and/or CB2 receptor antagonists.Assessing the therapeutic potential of CB1 and/or CB2 receptor allosteric modulators and neutral antagonists.CB1 and/or CB2 receptor allosteric modulators and neutral antagonists modulate for or against the presence of endocannabinoid mechanisms in mammalian cells.Validating and characterizing non-CB1 and non-CB2 targets such as RTK and TLR receptors for particular cannabinoids and developing compounds that can selectively activate or block such targets with reasonable potency for metabolic health and related diseases.Validating and characterizing that cannabinoid receptors may exist as homodimers or form heterodimers or oligomers with one or more classes of the non-cannabinoid receptor.Validating and characterizing the role played by the endocannabinoid system in ameliorating the symptoms and/or the underlying pathology of certain disorders.

## Figures and Tables

**Figure 1 cancers-15-01030-f001:**
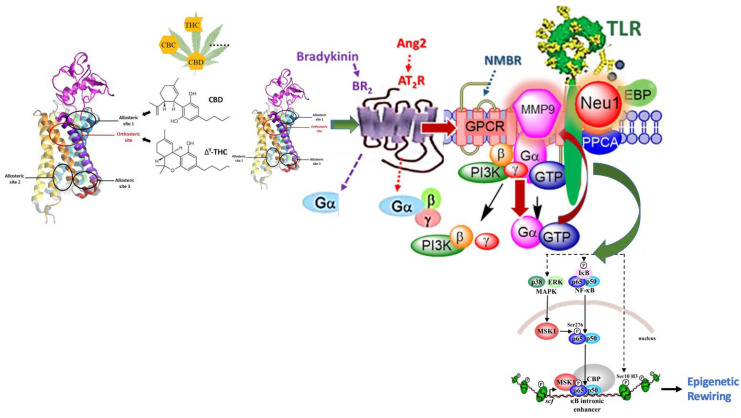
Bradykinin (BR2) and angiotensin II receptor type I (AT2R) are tethered within a multimeric receptor IRβ, receptor (TLR) transactivation-signaling axis, mediated by Neu1 sialidase and the glycosylation modification of TLRs. CB1 cannabinoid receptor with modeled *N*-terminus and the locations of orthosteric and allosteric binding sites. Cannabidiol (CBD)—An allosteric ligand, and Δ9-tetrahydrocannabinol (Δ9-THC)—an orthosteric ligand of CB1R. According to Lapraire et al., [96] and Sabatucci et al., [97], CBD ligands very likely bind to the allosteric site 1. The biased GPCR-signaling platform here potentiates Neu1 and MMP-9 cross-talk on the cell surface, which is essential for the transactivation of TLRs and subsequent NFκB cellular signaling and inducing epigenetic rewiring. Notes: TLR ligand, as well as GPCR agonists, can potentiate biased NMBR-TLR signaling and subsequently induce MMP-9 activation and Neu1 sialidase activity. Activated MMP-9 is proposed here to remove the EBP as part of the molecular multienzymatic complex that contains β-galactosidase/Neu1 and PPCA. Activated Neu1 then hydrolyzes α-2,3 sialyl residues of TLR at the ectodomain to remove steric hindrance to facilitate TLR association and subsequent recruitment of MyD88 and downstream signaling. Citation: Taken in part from Qorri et al., (2018) [98], Jakowiecki et al., (2021) [99], (The article is open access article distributed under the terms and conditions of the Creative Commons Attribution (CC BY) license (http://creativecommons.org/licenses/by/4.0/ (accessed on 23 April 2021)) and Reber et al., (2009) [100]. (This is an open-access article distributed under the terms of the Creative Commons Attribution License, which permits unrestricted use, distribution, and reproduction in any medium, provided the original author and source are properly credited).

**Figure 2 cancers-15-01030-f002:**
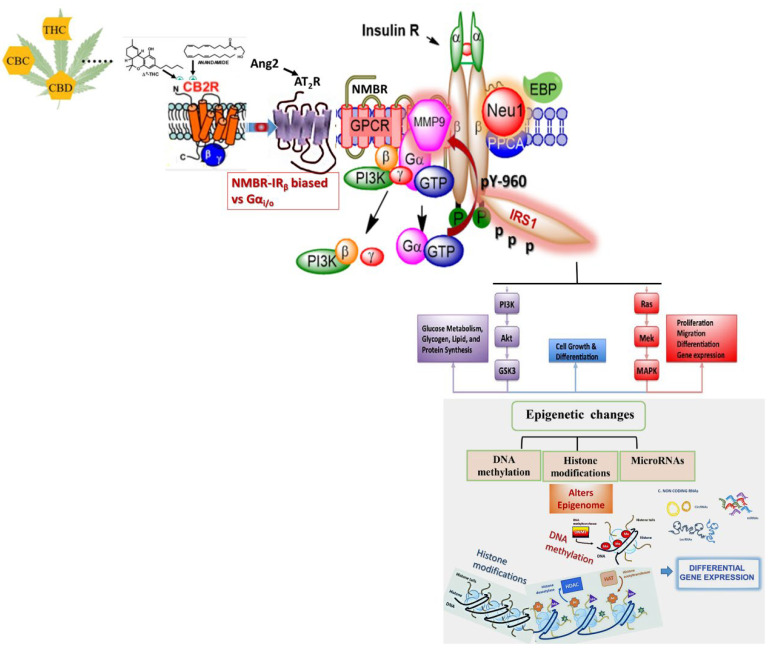
Angiotensin II receptor type I (AT2R) forms a multimeric receptor complex with neuromedin B GPCR (NMBR), insulin receptor β subunit (IRβ), and neuraminidase-1 (Neu1) in naïve (unstimulated) and stimulated HTC-IR cells [105]. Here, the interaction and signaling mechanism(s) of these molecules uncover a biased GPCR agonist-induced IRβ trans-activation signaling axis, mediated by Neu1 sialidase with the modification of insulin receptor glycosylation. The biased G protein-coupled receptor (GPCR)-signaling platform potentiates Neu1 and matrix metalloproteinase-9 (MMP-9) cell surface cross-talk that is essential for the activation of the IRβ tyrosine kinases. Notes: Insulin-binding receptor α subunits (IRα) as well GPCR agonists potentiate a biased NMBR-IRβ signaling and MMP-9 activation to induce Neu1 sialidase. Activated MMP-9 is proposed to remove the elastin-binding protein (EBP) as part of the molecular multi-enzymatic complex that contains β-galactosidase/Neu1 and protective protein cathepsin A (PPCA). Activated Neu1 hydrolyzes α-2,3 sialyl residues of IRβ to remove steric hindrance to facilitate IRβ subunits association and tyrosine kinase activation. Activated phospho-IRβ subunits phosphorylate insulin receptor substrate-1 (pIRS1), which initiates intracellular insulin signaling via the Ras-MAPK and the PI3K-Akt pathway, among others, such as NFκB. Abbreviations: PI3K: phosphatidylinositol 3-kinase; GTP: guanine triphosphate; IRS1: insulin receptor substrate-1; p: phosphorylation. Citation: Taken in part from Haxho et al., (2018) [105], Alghamdi et al., (2014) [112], Bordoni et al., (2019) [113], 2019 Elsevier B.V. and Société Française de Biochimie et Biologie Moléculaire (SFBBM). Open access under CC BY-NC-ND license. This is an Open Access article which permits unrestricted non-commercial use, provided the original work be properly cited.

**Figure 3 cancers-15-01030-f003:**
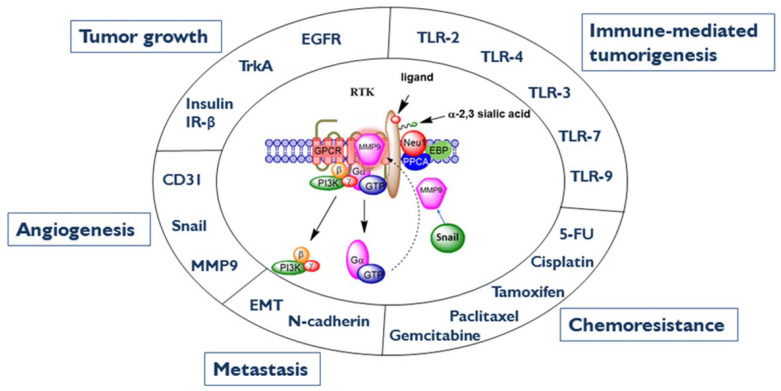
Neu1-MMP9-GPCR signaling platform in the regulation of RT K and the molecular targeting of multistage tumorigenesis. Neuraminidase-1 (Neu1) and matrix metalloproteinase-9 (MMP9) cross-talk in alliance with G protein-coupled receptor(s) (GPCR) regulates receptor tyrosine kinases (RTKs) and extracellular and intracellular Toll-like (TLR) receptors in cancer cells. This process sets the stage for multistage tumorigenesis. Citation: Taken in part from: Abdulkhalek et al., (2013) [118], Abdulkhalek et al., (2014) [134]. Publisher and licensee Dove Medical Press Ltd. These are an Open Access articles which permits unrestricted non-commercial use, provided the original work is properly cited.

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
