# Peer review of "Cannabinoids Transmogrify Cancer Metabolic Phenotype via Epigenetic Reprogramming and a Novel CBD Biased G Protein-Coupled Receptor Signaling Platform"

_cancers, 2023, doi:10.3390/cancers15041030_

Round 1

Reviewer 1 Report

A. Summary: Bunsick and colleagues presented a review on how endocannabinoids’ binding to cannabinoid receptor ( CB2) can activate receptor tyrosine kinases and Toll-like receptors (with the involvement of biased GPCR-Neu1-MMP9 signaling axis),  which may then lead to downstream activation of  NF-kB and the induction of epigenetic (DNA methylation, histone modifications, and RNA-associated alterations) and metabolic reprogramming, towards cancer proliferation or  metastasis. The assumption is that antagonists (e.g.  CBD), or  inhibition of some of the components of the  biased GPCR-Neu1-MMP9 signaling axis may prevent or slow down this process.

B. Comments:

1. The manuscript is comprehensively written. However, for those who are not very familiar with this particular topic, some aspects of the manuscript can be confusing. Therefore, I recommend including a Figure that illustrates the main points of the manuscript. The figure encompasses the cannabinoids (including antagonists)  binding to the GPCR receptors, the downstream processes, their effects on epigenetic processes, and the expression of genes with roles in cancer progression (apoptosis, proliferation, metastasis). In other words, please present a model illustrating why targeting Neu1, MMP9 (and other targets within the model),  or adding CB2 antagonists (perhaps CBD ) may reverse cancer phenotypes. It should be an expanded version of your current figure for the graphical abstract. A recent paper by Domingos et al. (“Regulation of DNA Methylation by Cannabidiol and Its Implications for Psychiatry: New Insights from In Vivo and Silico Models) might help.

2. I suggest a simpler synonym for “transmogrify.”

Author Response

Reviewer # 1

Comments and Suggestions for Authors

  1. Summary: Bunsick and colleagues presented a review on how endocannabinoids’ binding to cannabinoid receptor (CB2) can activate receptor tyrosine kinases and Toll-like receptors (with the involvement of biased GPCR-Neu1-MMP9 signaling axis), which may then lead to downstream activation of NF-kB and the induction of epigenetic (DNA methylation, histone modifications, and RNA-associated alterations) and metabolic reprogramming, towards cancer proliferation or  metastasis. The assumption is that antagonists (e.g.  CBD), or inhibition of some of the components of the  biased GPCR-Neu1-MMP9 signaling axis may prevent or slow down this process.  
  2. Comments:
  3. The manuscript is comprehensively written. However, for those who are not very familiar with this particular topic, some aspects of the manuscript can be confusing. Therefore, I recommend including a Figure that illustrates the main points of the manuscript. The figure encompasses the cannabinoids (including antagonists) binding to the GPCR receptors, the downstream processes, their effects on epigenetic processes, and the expression of genes with roles in cancer progression (apoptosis, proliferation, metastasis). In other words, please present a model illustrating why targeting Neu1, MMP9 (and other targets within the model), or adding CB2 antagonists (perhaps CBD ) may reverse cancer phenotypes. It should be an expanded version of your current figure for the graphical abstract. A recent paper by Domingos et al. (“Regulation of DNA Methylation by Cannabidiol and Its Implications for Psychiatry: New Insights from In Vivo and Silico Models) might help. 

Author response: Thank you for these comments and suggestions. We have included two Figures illustrating how CBD GPCR transactivates TLR and RTK receptors with subsequent downstream signaling and epigenetic rewiring. We have also provided evidence from the literature that this novel CBD GPCR biased signaling paradigm might be acting transactivation of these receptors depending on biased GPCR agonism, which refers to the ability of GPCR ligands to activate distinct intracellular signaling pathways by preferentially stabilizing different active conformational states of the receptor [92]. Both orthosteric ligands, such as THC, and allosteric ligands, such as CBD, tend to induce biased signaling at GPCRs [92]. Importantly, biased agonism and allosteric modulation have been observed at the CB1 receptor; however, to our knowledge, biased modulation of NF-kB signaling has yet to be studied in the context of CB receptors [92]. Cannabinoids may induce epigenetic modifications through the activation of these glycosylated receptors and the subsequent initiation of NF-κB; however, cannabinoids do not activate these receptors directly [93].

  1. Khajehali, E.; Malone, D.T.; Glass, M.; Sexton, P.M.; Christopoulos, A.; Leach, K. Biased agonism and biased allosteric modulation at the CB1 cannabinoid receptor. Mol Pharmacol 2015, 88, 368–379.
  2. Olianas, M.C.; Dedoni, S.; Onali, P. Cannabinoid CB1 and CB2 receptors differentially regulate TNF-α-induced apoptosis and LPA1-mediated pro-survival signaling in HT22 hippocampal cells. Life Sciences 2021, 276, 119407, doi:10.1016/j.lfs.2021.119407.
  3. I suggest a simpler synonym for “transmogrify.”

Author response: Thank you for this suggestion. Some common synonyms of transmogrify are convert, metamorphose, transfigure, transform, and transmute. While all these words mean "to change a thing into a different thing," transmogrify suggests a strange or preposterous metamorphosis (a change of the form or nature of a thing or person into a completely different one, by natural or other means).

            We published a review article, entitled “The Next-Generation of Combination Cancer Immunotherapy: Epigenetic Immunomodulators Transmogrify Immune Training to Enhance Immunotherapy,” by Mokhtari et al. Cancers 2021, 13, 3596. https://doi.org/10.3390/cancers13143596.

Reviewer 2 Report

although it is a good work but there are few suggestions that will increase the readibility of the work.

1. please update the references. many are too old. include recent studies.

2. if possible please add computational methods and drugs discovered for the said target.

3. conclusion should clearly define the future potential in terms of research gap and limitations of the previous works.

Author Response

Reviewer # 2

Comments and Suggestions for Authors

although it is a good work but there are few suggestions that will increase the readibility of the work.

  1. please update the references. many are too old. include recent studies.

Author response: Thank you for the suggestion. We added additional citations to highlight the important aspects of the review.

  1. if possible, please add computational methods and drugs discovered for the said target.

Author response: Thank you for the suggestion. We added three figure highlighting method and drugs.

  1. conclusion should clearly define the future potential in terms of research gap and limitations of the previous works.

Author response: Thank you for the suggestion. We have added the following at the end of the review:

  1. Future research directions and limitations

                        The gaps in our knowledge regarding how CB cannabinoids are implicated in host metabolism via GPCR receptors reinforces the importance of research needed to understand the mechanistic action of these drugs on the body as promising metabolic candidates. The specific mechanisms and details of the effects of CB cannabinoids on host metabolism and energy homeostasis still remain to be elucidated. It is likely that future research in the area of cannabinoid pharmacology will be directed at:

  • exploring the structure-activity relationships of ligands that target the CB1 allosteric site or that behave as neutral CB1 and/or CB2 receptor antagonists.
  • assessing the therapeutic potential of CB1 and/or CB2 receptor allosteric modulators and neutral antagonists.
  • CB1 and/or CB2 receptor allosteric modulators and neutral antagonists modulate for or against the presence of an endocannabinoid mechanisms in mammalian cells.
  • validating and characterizing non-CB1, non-CB2 targets such as RTK and TLR receptors for particular cannabinoids and developing compounds that can selectively activate or block such targets with reasonable potency for metabolic health and related diseases.
  • validating and characterizing that cannabinoid receptors may exist as homodimers or form heterodimers or oligomers with one or more classes of non-cannabinoid receptor.
  • validating and characterizing the role played by the endocannabinoid system in ameliorating the symptoms and/or the underlying pathology of certain disorders.

Round 2

Reviewer 1 Report

The authors revised the manuscript as suggested by this reviewer. The current version  now acceptable for publication.